# Risk Factors for Mortality in Patients with *Elizabethkingia* Infection and the Clinical Impact of the Antimicrobial Susceptibility Patterns of *Elizabethkingia* Species

**DOI:** 10.3390/jcm9051431

**Published:** 2020-05-12

**Authors:** Hye Seong, Jung Ho Kim, Jun Hyoung Kim, Woon Ji Lee, Jin Young Ahn, Nam Su Ku, Jun Yong Choi, Joon Sup Yeom, Young Goo Song, Su Jin Jeong

**Affiliations:** 1College of Medicine, Yonsei University, Seoul 03722, Korea; shininghye@yuhs.ac (H.S.); QETU1111@yuhs.ac (J.H.K.); JUNHY138@yuhs.ac (J.H.K.); LEEWJ86@yuhs.ac (W.J.L.); comebacktosea@yuhs.ac (J.Y.A.); smileboy9@yuhs.ac (N.S.K.); seran@yuhs.ac (J.Y.C.); JOONSUP.YEOM@yuhs.ac (J.S.Y.); IMFELL@yuhs.ac (Y.G.S.); 2Division of Infectious Diseases, Department of Internal Medicine, Korea University Guro Hospital, Seoul 08308, Korea; 3Division of Infectious Diseases, Department of Internal Medicine and AIDS Research Institute, Seoul 03722, Korea

**Keywords:** *Elizabethkingia*, risk factors, mortality, anti-bacterial agent, microbial sensitivity tests

## Abstract

*Elizabethkingia* species (spp.), which can colonize hospital environments, are emerging nosocomial pathogens presenting high mortality. Due to their intrinsic resistance to a broad range of antibiotics, optimal antibiotic dosage has yet to be determined against infections caused by *Elizabethkingia* spp. This study aimed to investigate the risk factors for the mortality of infections caused by *Elizabethkingia* spp. and assess the clinical implications of their antimicrobial susceptibility patterns. Data from 210 patients affected by *Elizabethkingia*-induced pneumonia and bacteremia between 1 November 2005 and 31 May 2016, were analyzed. Further antimicrobial susceptibility tests for moxifloxacin, rifampin, and vancomycin using *Elizabethkingia* isolates were performed to compensate for the *Elizabethkingia* spp. susceptibility panel in patients affected after 2013. The mean age of the patients was 66.5 ± 18 years and the 28-day mortality rate was 25.2% (53/210). In the univariate analysis, history of prior stay in an intensive care unit, central venous catheter use, presented thrombocytopenia, immunocompetent status, a high simplified acute physiology score II (SAPS II score), a high C-reactive protein (CRP)/albumin ratio on the day of isolation and seven days later, and a high minimum inhibitory concentration (MIC) value of rifampin were significantly associated with a higher mortality rate. In the multivariate logistic regression analysis, the MIC values of rifampin (odds ratio (OR): 1.045; 95% confidence interval (CI): 1.006–1.085; *p* = 0.023), SAPS II score (OR: 1.053; 95% CI: 1.022–1.084; *p* = 0.001), and initial CRP/albumin ratio (OR: 1.030; 95% CI: 1.009–1.051; *p* = 0.004) were significantly associated with 28-day mortality. To reduce the mortality associated with *Elizabethkingia* infections, prediction of the clinical course using initial CRP/albumin ratio and SAPS II and early intervention are essential. Rifampin is a promising candidate as the drug of choice in treating *Elizabethkingia* infections.

## 1. Introduction

*Elizabethkingia* species (spp.) are non-fermentative, non-motile, oxidase-positive, and non-glucose-fermenting Gram-negative aerobic bacilli [1,2]. They are ubiquitous saprophytes found in freshwater, saltwater, and soil environments [3]. They can also survive in chlorine-treated municipal water supplies. Colonized sink basins and taps are potential reservoirs of *Elizabethkingia* spp. infection in hospital settings [4,5]. *Elizabethkingia* spp. does not constitute human microflora [6]; however, it may colonize patients via fluid contaminated medical devices (e.g., respirators, intubation tubes, humidifiers, incubators for newborns, and even antiseptic and saline solutions [4,7,8,9]. It can spread by wet and dry materials and surfaces, including the hands of hospital staff [10].

Six species are assigned to Elizabethkingia genus, Elizabethkingia meningoseptica, Elizabethkingia miricola [1], Elizabethkingia anophelis [11], Elizabethkingia bruuniana, Elizabethkingia ursingii, and Elizabethkingia occulta [12], with the first three considered to be medically important [13]. They generally possess low virulence, and thus, positive clinical cultures usually represent colonization. Hsu et al. (2012) reported that among E. meningoseptica isolated subjects, approximately half of adults and one-third of neonates were colonizers [14].

Two or three decades ago, *E. meningoseptica* was emerging as a nosocomial pathogen but nowadays, *E. anophelis* is more prevalent and likely to cause lethal opportunistic infections [15]. It can be explained why *E. meningoseptica* is currently the most well-known species. *Elizabethkingia* spp. can especially affect the patients that undergo chronic hemodialysis therapy, trauma victims with multiple injuries, patients undergoing medical and surgical interventions, and patients showing immunosuppression, prolonged hospital stay, use of an indwelling central venous catheter and other invasive medical devices, and exposure to multiple broad-spectrum antibiotics [16,17,18,19]. Previous researchers have established that this species is a dangerous opportunistic bacterial pathogen, causing meningitis, pneumonia, septic arthritis, osteomyelitis, endocarditis, conjunctivitis, cholangitis, urinary tract infection, cellulitis, abdominal infections, epididymitis, prolonged hospital bronchitis, sinusitis, dialysis-associated peritonitis, and prosthesis-associated joint infection in humans [13,14,20,21,22,23,24]. 

It is known that *Elizabethkingia* spp. has virulence factors including a propensity for biofilm formation [2], intracellular invasion [25], and chromosomal- [26] and plasmid-mediated [27] resistance to many antimicrobial drugs, including commonly used β-lactams [1]. For example, *E. meningoseptica* possesses two different types of β-lactamases (intrinsic class A extended-spectrum serine-β-lactamases and inherent class B metallo-β-lactamases), which makes it resistant to a broad range of antimicrobials that are routinely used for empiric treatment of infections caused by Gram-negative bacteria [26,28]. For that reason, *E. meningoseptica* is generally resistant to carbapenems, extended-spectrum cephalosporins, aminoglycosides, aztreonam, and colistin [29,30]. It is no wonder that *E. meningoseptica* has emerged as a very successful nosocomial pathogen causing severe life-threatening infections in intensive care units where there is selective antibiotic pressure due to the higher antibiotic use [16]. The mortality associated with *E. meningoseptica* infection is alarmingly high (33–52%) [18], especially in patients who have received inappropriate initial antimicrobial therapy [2,31].

So far, however, there is no antimicrobial regimen of choice for empiric treatment of *Elizabethkingia* spp. infections, as antimicrobial susceptibilities have been inconsistent across reports [32]. Interestingly, *E. meningoseptica* is often susceptible to agents generally used to treat infections caused by Gram-positive bacteria such as rifampicin, clindamycin, erythromycin, trimethoprim-sulfamethoxazole (TMP-SMX), quinolones, and vancomycin.

In this study, we analyzed subjects showing *Elizabethkingia* spp. infections clinically and microbiologically. We aimed to reveal the risk factors for mortality in patients showing infection caused by *Elizabethkingia* spp. and find the clinical impact of their antimicrobial susceptibility patterns.

## 2. Materials and Methods

### 2.1. Patients and Study Settings 

From November 2005 to May 2016, all cases of *Elizabethkingia*-isolated pneumonia and bacteremia were identified retrospectively from electronic medical records at the Severance Hospital, a major medical center, and tertiary teaching hospital in South Korea, with a capacity of more than 2000 beds. After excluding patients under 18 years of age, we analyzed the trend in clinicophysiologic parameters 48 h before and after sample acquisition to identify patients with confirmed *Elizabethkingia* infection. Since most patients were ventilated, three systemic inflammatory response syndrome parameters: new temperature change to <36 °C or >38 °C, a new increase in pulse rate to >90 beats per minute, and new change in leukocyte count to <4 or >12 × 10^9^ cells/L, were investigated. In addition, we investigated three additional criteria: a new rise in the fraction of inspired oxygen requirement >0.1, new C-reactive protein (CRP) > 100 mg/L, and new pulmonary infiltrates on plain chest radiography.

A total of 210 patients (186 pneumonia and 24 bacteremia patients) were enrolled. The demographic data, medical history, simplified acute physiology score II (SAPS II), underlying diseases, Charlson comorbidity score, clinical manifestations, vital signs, chest radiography, laboratory test results (data on the day of microbial isolation and seven days later), microbiological data, and treatment regimen from each patient were collected and reviewed. To assess the impact of *Elizabethkingia* infection, we evaluated the 28-day mortality.

### 2.2. Ethical Approval and Consent to Participate

The study was approved (4-2019-1264) by the institutional review board of the Yonsei University Health System Clinical Trial Center. The informed consent was waived because this study was a retrospective study with review of related data through the electronic medical records. 

Death occurring within 28 days after diagnosis of the *Elizabethkingia* infection was defined as 28-day mortality. An episode of bacteremia was defined as the presence of one or more blood cultures that were positive for *Elizabethkingia* spp., which contributed to clinical sepsis. An episode of pneumonia was defined as the presence of sputum or bronchial alveolar lavage cultures that were positive for *Elizabethkingia* spp. with infiltration in radiological imaging.

Prolonged hospitalization was defined as hospitalization for two or more weeks before the diagnosis of bacteremia. Prior stay in an intensive care unit was defined as a stay in the intensive care unit for more than seven days within 14 days prior to the diagnosis of bacteremia or pneumonia. Recent surgery was defined as a surgical procedure, excluding tracheostomy, performed within one month prior to the onset of *Elizabethkingia*-induced bacteremia or pneumonia. Immunosuppressive therapy was defined as the use of cytotoxic agents or corticosteroids (more than 30 mg of prednisolone daily or the equivalent for one week or more). 

Prolonged antibiotic use was defined as the administration of intravenous antibiotics for more than 14 days within 30 days before the diagnosis of bacteremia. Appropriate antibiotic therapy was defined as the use of at least one intravenous antibiotic to which the microorganism was susceptible, according to the corresponding minimum inhibitory concentration (MIC), within 72 h of diagnosis of the infection.

### 2.3. Bacterial Isolates and Antimicrobial Susceptibility Testing

Species identification initially identified using the Vitek II GN card system (bioMerieux, Marcy l’Etoile, France) but updated by two matrix-assisted laser desorption ionization time-of-flight systems, the Bruker Biotyper (Bruker Daltonics, Bremen, Germany) and the Vitek mass spectrometry (bioMerieux, Marcy l’Etoile, France) as described by Han et al. (2016) [33]. Strains with discrepant results were confirmed by 16S rRNA gene sequencing using universal primers [34].

Susceptibilities of the isolates to antimicrobials were determined using epsilometer test (E-test) strips (bioMerieux, Marcy l’Etoile, France). Sputum, bronchial alveolar lavage, and blood culture samples were processed using the BACTEC FX system (Becton Dickinson, Sparks, MD, USA). All positive cultures were examined by Gram staining and were sub-cultured on blood agar plates and eosin-methylene blue agar plates for further identification.

The following antimicrobial agents were assayed: ceftazidime, cefobactam, cefoxitin, cefotaxim, cefepime, aztreonam, piperacilin-tazobactam, amoxicillin-clavulinate, ampicillin/sulbactam, imipenem, meropenem, tobramycin, amikacin, gentamicin, erythromycin, minocycline, ciprofloxacin, levofloxacin, trimethoprim–sulfamethoxazole, teicoplanin, vancomycin, tigecycline, and colistin. With no established MIC breakpoints of *Elizabethkingia spp.* for moxifloxacin, the interpretive breakpoints of non-*Enterobacteriaceae* for gatifloxacin were used. For the reasons mentioned above, the MIC breakpoints for *Staphylococcus* spp. and *Staphylococcus aureus* were used to interpret the susceptibility of rifampin and vancomycin, respectively [35].

Since 2013, moxifloxacin, rifampin, and vancomycin have been excluded from the antibiotic susceptibility panel of *Elizabethkingia* spp. To compensate for the *Elizabethkingia* spp. susceptibility panel, further E-tests (bioMerieux, Marcy l’Etoile, France) for those antibiotics were carried out using 122 *Elizabethkingia* isolates, which were stored in tryptic soy broth with 15% glycerol at −70 °C.

### 2.4. Statistical Analysis

All variables are expressed as the mean ± standard deviation unless otherwise indicated. Continuous variables were compared with the Student *t*-test or the Mann–Whitney nonparametric test. The proportions were compared with the χ^2^ or Fisher exact test when appropriate. Logistic regression was used for multivariate analysis of the risk factors associated with 28-day mortality, and the results are presented as adjusted odds ratios (OR) with 95% confidence intervals (CI). Variables were selected for the multivariate analysis based on the level of significance of the univariate association with 28-day mortality (*p* < 0.05). Models were built sequentially, starting with the variable most strongly associated with 28-day mortality and continued until no other variable reached significance or altered the odds ratios of variables already in the model. Upon arriving at the final model, each variable was sequentially excluded to assess its effect. Statistical analyses were performed using the Statistics Package for Social Science (SPSS 25.0 for windows; SPSS Inc., Chicago, IL, USA). *P*-values less than 0.05 were considered statistically significant.

## 3. Results

### 3.1. Population Characteristics

From November 2005 to May 2016, the bacterial cultures from 492 patients yielded *Elizabethkingia* spp. Among them, 43 patients aged under 18 years, 217 patients showing colonization with *Elizabethkingia* spp., and 22 patients presenting with other infections (except pneumonia and bacteremia) were excluded. Finally, 210 patients (186 pneumonia patients and 24 bacteremia patients) with *Elizabethkingia* infection were enrolled during the study periods.

Table 1 summarizes the baseline characteristics of the 210 patients. The mean age of the patients was 66.5 ± 18 years, and the 28-day mortality rate was 25.2% (53/210). The mean duration from admission to the isolation of *Elizabethkingia* spp. was 55.8 ± 74.9 days. The most common underlying disease was malignancy (*n* = 100), followed by cerebrovascular attack (*n* = 47). The majority of subjects who contracted *Elizabethkingia* infection had pneumonia (*n* = 186).

The survivors had a lower SAPS II score (43.5 ± 20.2 vs. 59.9 ± 19.8, *p* < 0.001) and a higher incidence of concomitant pathogens (80 (51.0%) vs. 17 (32.1%), *p* = 0.017) than the non-survivors (Table 1). Charlson comorbidity index, prolonged hospitalization, and underlying disease were not statistically different between the two groups. 

Table 2 describes the clinical conditions and laboratory data among patients with *Elizabethkingia* infection. In comparison with survivors, the non-survivors presented a higher percentage with regard to the following aspects: previous intensive care unit (ICU) admission (40 (75.5%) vs. 89 (56.7%), *p* = 0.015), central venous catheter use (48 (90.6%) vs. 121 (77.1%), *p* = 0.032), and immunocompromised status (24 (45.3%) vs. 42 (26.8%), *p* = 0.012). Non-survivors who reported a high rate of thrombocytopenia (34 (64.2%) vs. 47 (29.9%), *p* < 0.001) and CRP/albumin ratio (50.4 ± 37.2 vs. 22.6 ± 26.9, *p* < 0.001) on the day of isolation also reported significantly higher rates of thrombocytopenia (23 (63.9%) vs. 38 (26.0%), *p* < 0.001) and CRP/albumin ratio (45.7 ± 30.1 vs. 18.2 ± 18.9, *p <* 0.001) seven days later. In contrast to these parameters, hypoalbuminemia was remarkable on the days of isolation (34 (64.2%) vs. 64 (41.0%), *p* = 0.004), but was not significant seven days later (20 (57.1%) vs. 61 (42.1%), *p* = 0.108).

### 3.2. Antibiogram Patterns and Therapeutic Regimens 

The list of isolated *Elizabethkingia* species, their susceptibility to antibiotics, and the MICs of the antimicrobial agents for these species are shown in Table 3. *E. meningoseptica* was the most common pathogenic member among *Elizabethkingia* species (*n* = 132), and its contribution to mortality was similar to that of the other species. The results of E-tests for moxifloxacin, rifampin, and vancomycin from 122 preserved *Elizabethkingia* isolates were added to the previously examined data of the susceptibilities of *Elizabethkingia* spp. For the non-survivor group, the antibiotic susceptibility rate was found to be low for rifampin and vancomycin, and high for moxifloxacin, but no statistical significance was seen. In addition, the MIC value of rifampin in the non-survivor group was significantly higher than that in the survivor group (11.9 ± 15.6 vs. 6.0 ± 11.7, *p =* 0.031). The proportion of isolated *Elizabethkingia* species and their susceptibility rate to antimicrobial agents are shown in Figure 1. Regardless of the type of *Elizabethkingia* species, both moxifloxacin and vancomycin were active against these pathogens. However, rifampin, which was related to the mortality associated with *Elizabethkingia* infections, showed various susceptibilities and was the most active against *E. miricola.*

The treatment duration and antimicrobial treatment regimens in the study population are described in Table 4. More than 80% of the patients had a history of prolonged antibiotics usage at the time of the isolation of *Elizabethkingia* spp. A majority of the therapeutic procedures involved a combination of antibiotics (*n* = 187). The antimicrobial regimen including glycopeptide was the most common (*n* = 127). No differences according to the antimicrobial regimen were observed between the survivor and non-survivor groups, except in case of the anti-pseudomonal penicillin plus glycopeptide regimen (20 (12.7%) vs. 2 (3.8%), *p* = 0.045).

### 3.3. Risk Factors for 28-day Mortality 

In the multivariate logistic regression analysis, the MIC of rifampin was associated with 28-day mortality (OR: 1.045; 95% CI: 1.006–1.085; *p* = 0.023). As the SAPS II increased, the mortality also increased (OR: 1.053; 95% CI: 1.022–1.084; *p* = 0.001). The initial CRP/albumin ratio also had an association with 28-day mortality (OR: 1.030; 95% CI: 1.009–1.051; *p* = 0.004).

## 4. Discussion

*Elizabethkingia* spp. is an emerging nosocomial pathogen. It is known that *E. meningoseptica* colonizes in human oropharynx [36], respiratory secretions [4], aerosol tubes [36], endotracheal tubes [8,36], and the respiratory tract in ventilated adult patients [37,38,39]. It has been found that *Elizabethkingia* spp. is not only a human colonizer, but also an opportunistic pathogen causing in-hospital outbreaks [18,39,40] resulting in high mortality [3,38].

A possible explanation for the high mortality rate may be its potential to form biofilms. The biofilms produced by biofilm-forming pathogens, like *Elizabethkingia* spp., might decrease the immune response and increase resistance against antibiotics, resulting in a high mortality rate [2]. Previous studies have presented that the cumulative mortality rate of *E. meningoseptica* infections was considerable in non-neonates (25%; [19], and 30%; [41]). The results of our study demonstrate that the 28-day mortality rate of *E. meningoseptica* infections was 25.2%, consistent with previously reported mortality.

In this study, we investigated the risk factors for mortality in patients with *Elizabethkingia* spp. infection. According to the multivariate logistic regression analysis, higher MIC of rifampin, increased SAPS II, and elevated CRP/albumin ratio on the day of bacteremia were significantly related to mortality. Previous studies have demonstrated that severe underlying diseases [42], prolonged hospitalization, treatment with invasive procedures, prior use of broad-spectrum antimicrobials, concomitant infections [35], and ICU stay [14] impacted survival rates. Moreover, Lin et al. reported that the usage of intravenous catheters was a predictor of fatal outcomes [31]. Since an increased SAPS II and elevated CRP/albumin ratio usually reflect disease severity, our results are in line with those of previous studies. The early identification of these clinical factors in patients with *Elizabethkingia* infection could be essential to improve prognosis.

Several studies have reported that a lack of treatment with effective antibiotics was an independent predictor of mortality [14,31,42]. Contrary to our expectations, this study did not find a significant difference between survivors and non-survivors with regard to effective antibiotic use. A possible explanation for this might be that other co-pathogens could make it difficult to differentiate the impact of *Elizabethkingia* spp. and may reflect the influence of other co-pathogens on the mortality rates; about half of the patients in the study showed infection with at least two pathogens. Interestingly, we found that the survivors had a higher incidence of co-pathogens than the non-survivors. It might be suggested that *Elizabethkingia* may not act as a true pathogen when they mixed with other pathogens.

The most important clinically relevant finding in our study is that the elevated MIC value of rifampin was an independent risk factor associated with the mortality of *Elizabethkingia* infections. As the MIC value of rifampin increased, the mortality rate associated with *Elizabethkingia* infections increased. It might be explained by mutations in RNA polymerase. RNA polymerase mutations not only induce rifampin resistance but also cause other alterations, especially virulence-related factors. For instance, *Staphylococcus aureus*, one of the biofilm-forming bacteria like *Elizabethkingia*, which gained RNA polymerase mutations showing a high level of rifampin MIC values, presented the elevated capacity for biofilm formation [43].

Owing to its marked intrinsic resistance, particularly to β-lactam antibiotics [18,38,42,44,45], there has been no consensus on the drug of choice for *E. meningoseptica* infection. One interesting finding was that paradoxically, *E. meningoseptica* presented sensitivity to antibiotics that have effects against Gram-positive bacteria, such as fluoroquinolones, rifampin, TMP-SMX, and vancomycin [9,35]. It is interesting to note that this research found that *Elizabethkingia* spp. presented high susceptibility to rifampin, moxifloxacin, and vancomycin. These results are consistent with those observed in earlier studies.

A recent study showed that three *Elizabethkingia* spp. (*E. meningoseptica*, 100%; *E. miricola*, 94.4%; *E. anophelis*, 94.4%) have high susceptibility to rifampin [46]. Due to its higher susceptibility and relevance to mortality, rifampin could be a promising candidate for a treatment regimen against *Elizabethkingia* spp. infections. Previous studies also suggest that rifampin [40], moxifloxacin [9,14,29,47], minocycline [38,42,48], TMP-SMX alone [49] or in combination with ciprofloxacin [50], and anti-pseudomonal penicillins plus amikacin [51] could be considered as potential treatment strategies for *Elizabethkingia* spp. infections. Since 2013, the antibiotics susceptibility panel for *Elizabethkingia* spp. in our hospital did not include rifampin; hence, clinical data of rifampin usage for *Elizabethkingia* seldom exists. Further research investigating the appropriate antibiotic regimens, including those of rifampin, is essential to reduce the risk of mortality of *Elizabethkingia* infection.

An E-test for moxifloxacin demonstrated that it remained an active agent against all three *Elizabethkingia* spp. However, there were no significant differences in the mortality between groups that used moxifloxacin and those that did not. It might result from the bacteria’s several mechanisms facilitating resistance include acquisition of efflux pumps or protection enzyme encoding plasmids that inactivate the fluoroquinolone [52].

Pereira et al. (2013) recommended vancomycin as a drug of choice for neonatal meningitis caused by *E. meningoseptica* [35]. Chang et al. (2019) reported the low resistance of *Elizabethkingia* spp. against vancomycin (*E. meningoseptica*, 0%; *E. miricola*, 21.4%; *E. anophelis*, 4.2%) [46]. In our study, however, no significant correlation was found between vancomycin usage and mortality following *Elizabethkingia* infection. This inconsistency may be due to the discrepancy of the target of vancomycin usage. Usually, vancomycin is used against Gram-positive bacteria. *Elizabethkingia* species are Gram-negative, hence, the purpose of vancomycin usage was perhaps, not to cure *Elizabethkingia* infection, but instead control the growth of other Gram-positive bacteria.

It is interesting to note that in univariate analysis, survivors of this study showed efficacy not for anti-pseudomonal penicillin or vancomycin alone, but anti-pseudomonal penicillin plus glycopeptide regimen. These may result reflect those of Di Pentima et al. (1998) who also found that in vitro vancomycin synergy against *Elizabethkingia meningoseptica* (originally named *Flavobacterium meningoseptica*) [53,54]. Considering its resistant mechanism, glycopeptides that disrupt the bacterial membrane could make it easy for beta-lactams to penetrate the bacteria resulting in favorable outcomes [52]. It can thus be suggested that in severe *Elizabethkingia*-infected patients, vancomycin accompanied by rifampin or anti-pseudomonal penicillin can be promising candidates unless their antibiotics susceptibility test presented resistance. There are several limitations to our study. First, our study is retrospective. The history of antibiotics usage was dependent on electronic medical records and no intervention was performed with regard to the choice of antibiotics. Secondly, our study population was quite small to determine the risk factors affecting mortality. Finally, we summarized three *Elizabethkingia* spp. to analyze the risk factors for mortality. The mortality rates for infections caused by each species could be different. However, upon checking for differences in the mortality based on the type of species, we found that the mortalities of the infections caused by the three species were similar. Despite these limitations, our study revealed that rifampin is the most active agent against the *Elizabethkingia* spp. Further randomized controlled trials with more subjects need to be undertaken to support our results.

## 5. Conclusions

We conducted this study to identify the risk factors for the mortality associated with *Elizabethkingia* infections and to find the clinical impacts of the antimicrobial susceptibility patterns of *Elizabethkingia* spp., which are emerging nosocomial pathogens. Our results suggest that the MIC values of rifampin, SAPS II score, and initial CRP/albumin ratio were significantly related to the 28-day mortality associated with infections caused by *Elizabethkingia* spp.

Prediction of patients’ clinical courses using initial CRP/albumin ratio and SAPS II is a priority to reduce the mortality caused by *Elizabethkingia* infections. Rifampin is a promising candidate as the drug of choice for *Elizabethkingia* infection. Therefore, to reduce the mortality, the use of antibiotics in *Elizabethkingia*-infected patients based on these predictors should be carefully considered.

## Figures and Tables

**Figure 1 jcm-09-01431-f001:**
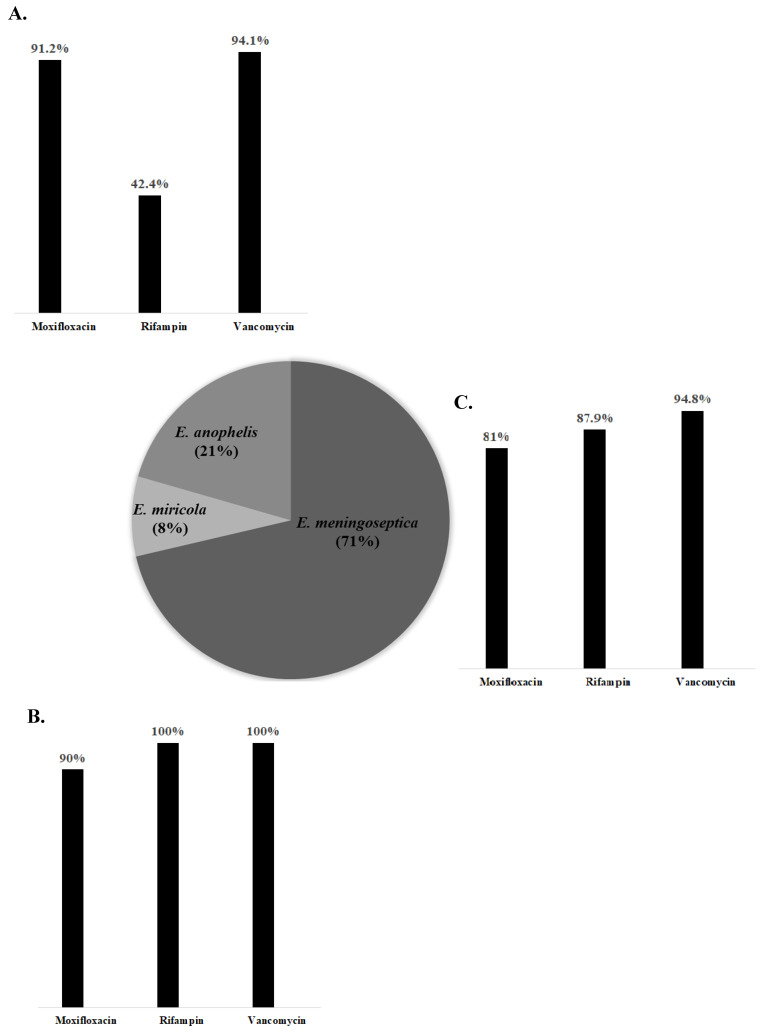
The proportion of isolated *Elizabethkingia* species and in vitro antibiotic susceptibility rate of three *Elizabethkingia* species. The proportions of three *Elizabethkingia* species are presented in a pie chart, and the percentage of each species is shown in parentheses. The susceptibilities of each *Elizabethkingia* species to three antibiotics (moxifloxacin, rifampin, and vancomycin) are also presented in bar charts. (**A**), *E. anopheles*, (**B**), *E. miricola*, and (**C**), *E. meningoseptica,* respectively.

**Table 1 jcm-09-01431-t001:** Baseline characteristics of 210 patients with *Elizabethkingia spp*. infection.

Characteristics	Total (*n* = 210)	Survivor (*n* = 157)	Non-Survivor (*n* = 53)	*p*-Value
**Sex (male)**	129 (61.4)	98 (62.4)	31 (58.5)	0.730
**Age (years)**	66.5 ± 18	63.7 ± 16.5	62.5 ± 14.1	0.637
**HOD (days)**	55.8 ± 74.9	58.6 ± 83.2	47.6 ± 41.2	0.357
**Charlson comorbidity index**	3.3 ± 2.1	3.2 ± 2.2	3.5 ± 2.1	0.419
**SAPS II**	47.7 ± 21.3	43.5 ± 20.2	59.9 ± 19.8	<0.001
**Underlying disease, yes**				
** MI**	25 (11.9)	17 (10.8)	8 (15.1)	0.409
** CHF**	30 (14.3)	22 (14.0)	8 (15.1)	0.846
** PAOD**	9(4.3)	7 (4.5)	2 (3.8)	0.831
** Cerebrovascular attack**	47 (22.4)	39 (24.8)	8 (15.1)	0.141
** Dementia**	10 (4.8)	8 (5.1)	2 (3.8)	0.696
** Chronic pulmonary disease**	43 (20.5)	34 (21.7)	9 (17.0)	0.466
** Connective tissue disease**	7 (3.3)	5 (3.2)	2 (3.8)	0.836
** Liver disease**	21 (10.0)	14 (8.9)	7 (13.2)	0.427
** Diabetes mellitus**	41 (19.5)	32 (20.4)	9 (17.0)	0.589
** Hemiplegia**	14 (6.7)	13 (8.3)	1 (1.9)	0.198
** CKD**	41 (19.5)	30 (19.1)	11 (20.8)	0.794
** ESRD**	28 (13.3)	21 (13.5)	7(13.2)	0.963
** Malignancy**	100 (47.6)	73 (46.5)	27 (50.9)	0.634
** Transplantation**				
** SOT**	17 (8.1)	12 (7.6)	5 (9.4)	0.771
** PBSCT**	8 (3.8)	6 (3.8)	2 (3.8)	>0.999
**Polymicrobial infection**	97 (46.2)	80 (51.0)	17 (32.1)	0.017
**Source of infection**				0.304
** Pneumonia**	186 (88.6)	137 (87.3)	49 (92.5)	
** Primary bacteremia**	24 (11.4)	20 (12.7)	4 (7.5)	

HOD, hospital days of acquisition; SAPS II, simplified acute physiology score II; MI, myocardial infarction; CHF, congestive heart failure; PAOD, peripheral artery occlusive disease; CKD, chronic kidney disease; ESRD, end stage renal disease; SOT, Solid organ transplantation; PBSCT, Peripheral blood stem cell transplantation. Continuous variables are shown as the mean ± standard deviation (SD) and categorical variables, as numbers (percentage).

**Table 2 jcm-09-01431-t002:** Comparisons of clinical conditions and laboratory data between survivors and non-survivors.

Characteristics	Total (*n* = 210)	Survivor (*n* = 157)	Non-Survivor (*n* = 53)	*p*-Value
**Predisposing conditions, yes**				
** Prior stay in ICU**	129 (61.4)	89 (56.7)	40 (75.5)	0.015
** Recent surgery**	44 (21.0)	31 (19.7)	13 (24.5)	0.459
** Ventilator use**	149 (71.0)	107 (68.2)	42 (79.2)	0.124
** Multiple injuries**	4 (1.9)	4 (2.5)	0(0.0)	0.574
**Immunocompromised status, yes**	66 (31.4)	42 (26.8)	24 (45.3)	0.012
**Invasive device use, yes**				
** Central venous catheter**	169 (80.5)	121 (77.1)	48 (90.6)	0.032
** Foley catheter**	178 (85.2)	130 (83.3)	48 (90.6)	0.201
** Endotracheal tube**	163 (77.6)	121 (77.1)	42 (79.2)	0.743
** Port-A catheter**	6 (2.9)	4 (2.5)	2 (3.8)	0.644
** Cerebrospinal shunt**	5 (2.4)	4 (2.5)	1 (1.9)	>0.999
** Pleural catheter**	36 (17.1)	24 (15.3)	12 (22.6)	0.219
** Pancreaticobiliary catheter**	8 (3.8)	5 (3.2)	3 (5.7)	0.419
**Laboratory test on Day 0**				
** Leukocytosis (>10,000 mm/µL)**	135 (64.3)	102(65.0)	33 (62.3)	0.722
** RDW (%)**	16.4 ± 2.5	15.7 ± 2.7	16.8 ± 4.0	0.221
** Delta-neutrophil (%)**	6.0 ± 20.1	1.7 ± 3.1	3.0 ± 6.0	0.727
** Thrombocytopenia, yes (/µL)**	81 (38.6)	47 (29.9)	34 (64.2)	<0.001
** Severe thrombocytopenia, yes (/µL)**	26 (12.4)	13 (8.3)	13 (24.5)	0.002
** Hypoalbuminemia, yes (g/dL)**	98 (46.9)	64 (41.0)	34 (64.2)	0.004
** CRP/albumin ratio**	26.6 ± 29.3	22.6 ± 26.9	50.4 ± 37.2	<0.001
** Renal insufficiency, yes (%)**	60 (28.6)	47 (29.9)	13 (24.5)	0.451
**Laboratory test on Day 7**				
** Leukocytosis (>10,000/µL)**	123 (67.6)	101 (69.2)	22 (61.1)	0.354
** RDW (%)**	16.7 ± 2.3	16.2 ± 2.4	16.6 ± 2.3	0.210
** Delta-neutrophil (%)**	3.7 ± 7.7	5.1 ± 19.7	6.3 ± 13.2	0.072
** Thrombocytopenia, yes (/µL)**	61 (33.5)	38 (26.0)	23 (63.9)	<0.001
** Severe thrombocytopenia, yes (/µL)**	21 (11.5)	11 (7.5)	10 (27.8)	0.001
** Hypoalbuminemia, yes (g/dL)**	81(45.0)	61 (42.1)	20 (57.1)	0.108
** CRP/albumin ratio**	24.3 ± 25.0	18.2 ± 18.9	45.7 ± 30.1	<0.001
** Renal insufficiency, yes (%)**	52 (28.6)	43 (29.5)	9 (25.0)	0.596

ICU, Intensive Care Unit; RDW, Red Cell Distribution Width; CRP, C-reactive protein of acquisition. Renal insufficiency was defined as a calculated glomerular filtration rate (GFR) of less than 60 mL per minute/1.73 m^2^. Thrombocytopenia and severe thrombocytopenia were defined as a platelet count of less than 100,000 and 50,000 per µL, respectively. Hypoalbuminemia was defined as a low blood level of albumin of less than 2.8 g per dL. Continuous variables are shown as the mean ± standard deviation (SD) and categorical variables, as numbers (percentage).

**Table 3 jcm-09-01431-t003:** Antibiogram patterns of *Elizabethkingia spp*. infection.

Antibiotics	Total (*n* = 210)	Survivor (*n* = 157)	Non-Survivor (*n* = 53)	*p*-Value
***Elizabethkingia* (%)**				0.513
*** E. meningoseptica***	132 (62.9)	94 (59.9)	38 (71.7)	0.123
*** E. anophelis***	38 (18.1)	30 (19.1)	8 (15.1)	0.512
*** E. miricola***	15 (7.1)	12 (7.6)	3 (5.7)	0.765
*** Species not identified***	25 (11.9)	21 (13.4)	4 (7.5)	0.257
**Susceptibility, yes (%)**				
** Minocycline**	186 (89.4)	138 (89.0)	48 (90.6)	0.805
** Cotrimoxazole**	150 (72.1)	116 (77.3)	34 (64.2)	0.157
** Anti-pseudomonal penicillins**	18 (8.7)	13 (8.4)	5 (9.4)	0.782
** Moxifloxacin**	103 (84.4)	76 (82.6)	27 (90.0)	0.400
** Rifampin**	92 (76.0)	73 (80.2)	19 (63.3)	0.060
** Vancomycin**	134 (96.4)	102 (95.3)	32 (23.9)	0.589
**MIC (ug/mL)**				
** Moxifloxacin (range: 0.012 to >32)**	2.0 ± 5.7	2.1 ± 5.7	1.5 ± 5.9	0.636
** Rifampin (range: 0.008 to >32)**	7.5 ± 13.0	6.0 ± 11.7	11.9 ± 15.6	0.031
** Vancomycin (range: 0.050 to >256)**	12.2 ± 39.4	14.6 ± 45.1	4.6 ± 3.1	0.230

MIC, minimal inhibitory concentration. Continuous variables are shown as the mean ± SD (standard deviation) and categorical variables, as numbers (percentage).

**Table 4 jcm-09-01431-t004:** Antimicrobial treatment regimen for 210 patients with *Elizabethkingia spp.* infection.

Antibiotics	Total (*n* = 210)	Survivor (*n* = 157)	Non-survivor (*n* = 53)	*p*-Value
**Prolonged antibiotics use (%)**	177 (84.3)	129 (82.2)	48 (90.6)	0.146
**Appropriate antibiotic therapy (%)**	92 (44.2)	70 (45.2)	22 (41.5)	0.644
**Antibiotics therapy**				0.803
** Monotherapy**	23 (11.0)	18 (11.5)	5 (9.4)	
** Combination therapy**	187 (89.0)	139 (88.5)	48 (90.6)	
**Antibiotics regimen**				
** Monotherapy**				
** Glycopeptide**	127 (60.5)	97 (61.8)	30 (56.6)	0.505
** Carbapenem**	124 (59.0)	90 (57.3)	34 (64.2)	0.382
** Fluroquinolone**	69 (32.9)	51 (32.5)	18 (34.0)	0.843
** Cephalosporin**	54 (25.7)	39 (24.8)	15 (28.3)	0.618
** Colistin**	54 (25.7)	36 (22.9)	18 (34.0)	0.112
** Anti-pseudomonal penicillin**	37 (17.6)	30 (19.1)	7 (13.2)	0.330
** Cotrimoxazole**	34 (16.2)	22 (14.0)	12 (22.6)	0.140
** Aminoglycoside**	16 (7.6)	10 (6.4)	6 (11.3)	0.242
**Combination therapy**				
** Carbapenem+glycopeptide**	94 (44.8)	72 (45.9)	22 (41.5)	0.582
** Carbapenem+colistin**	39 (18.6)	26 (16.6)	13 (24.5)	0.197
** Fluroquinolone+glycopeptide**	39 (18.6)	28 (17.8)	11 (20.8)	0.636
** Anti-pseudomonal penicillin+glycopeptide**	22 (10.5)	20 (12.7)	2 (3.8)	0.045
** Anti-pseudomonal penicillin+fluroquinolone**	10 (4.8)	8 (5.1)	2 (3.8)	>0.999
** Carbapenem+aminoglycoside**	9 (4.3)	6 (3.8)	3 (5.7)	0.695

Categorical variables are shown as numbers (percentage).

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
