# Peer review of "Risk Factors for Mortality in Patients with Elizabethkingia Infection and the Clinical Impact of the Antimicrobial Susceptibility Patterns of Elizabethkingia Species"

_jcm, 2020, doi:10.3390/jcm9051431_

Round 1
Reviewer 1 Report
Summary:
This manuscript presents a retrospective analysis of 210 patients with Pneumonia or Bacteremia due to infection with Elizabethkingia species. A variety of clinical parameters were evaluated to determine if they differed between survivors and non-survivors of the infection. The previous finding that early intervention with appropriate antibiotics has a large impact on survival was replicated, and certain previously unrecognized factors (such as initial CRP/albumin ratio) were found to predict or influence the clinical course. The comprehensive nature of the analysis showed that many other parameters (such as the Charlson comorbidity index) did not influence patient survival, and this is important information also. The discovery that Elizabethkingia isolates are widely susceptible to rifampicin but that MICs against this antibiotic are higher in non-survivors has direct clinical applicability.
Major Considerations:
- Much of the introduction reads like it was written three or four years ago, and not updated.
- Authors used a process for species identification that is known to be inaccurate. This would be a real problem if they were trying to assert differences between species. However, since no differences were observed, it should be fairly straightforward for them to revise to simply describe all infections as being due to bacteria identified to the genus level.
- Certain sentences are strangely structured; I’ve noted these in some cases. It would be worthwhile to have a native English speaker edit or co-author this manuscript.
Minor considerations:
Line 21: “colonize hospital environments;” should have a comma, not a semi-colon
Line 23: “ no appropriate antibiotics have yet been determined…” is a bit too strong of a statement; could be modified to “optimal antibiotic dosage has yet to be determined” or something like that.
Line 29: it is not clearly written what is lacking that needs to be compensated for
Line 54: “four species” …there are now six.
Line 56: “Elizabethkingia endophytica” now recognized as a subspecies of E. anophelis.
Line 57: “E. meningoseptica appears to be the most virulent species of Elizabethkingia;” this statement is unsupported and contradicts numerous papers in recent years that found E. anophelis to be more prevalent and likely to cause more serious infections.
Line 61: “necessitated chronic hemodialysis therapy”… Weird choice of words; why would someone be on hemodialysis if it were not necessary?
Line 64: “Previous researches have established…” Should be "Previous research has..." or "Previous researchers have..."
Line 74: “Owing to the presence of these enzymes…” The beginning of this long sentence is a complete statement, and it does not introduce the second part of the sentence. It should be its own sentence.
Line 81: “However, only a few studies…” needs elaboration. In what way are the studies that have been done already insufficient, beyond being few in number? Did the authors expect the results of their study to be different or more detailed?
Line 96: “After excluding patients under 18 years of age,…” Why were these excluded? It’s not a problem, just need more information.
Line 130: “Species identification and in vitro susceptibility testing of Elizabethkingia isolates were performed using the Vitek II…” Vitek II does not provide reliable Elizabethkingia species identification…see https://www.ncbi.nlm.nih.gov/pmc/articles/PMC6296276/
Line 221: “It is known that E. meningoseptica only 221 colonized in…” I’m puzzled by the use of "only" in this sentence; it is followed by a list of locations that have been shown to be colonized, but various publications are cited for each location, and it is not clear that other locations have been excluded as possibilities, or if they are talking about E. meningoseptica only, or what.
Line 227: “Infections by biofilm-forming pathogens…” It is the actual bio-film formation that leads to increased resistance, right? not simply the infection by an organism capable of filming biofilms? This sentence needs to be reworded for clarity.
Line 230: “…our study demonstrated…” This would be a good place to address how similar (or dis-similar) the population selected for this study was compared to previous findings, and what might account for the observed differences between study results.
Line 260: “…rifampicin has high susceptibility to three Elizabethkingia spp. …” Oddly worded. The bacteria has susceptibility to the antibiotic, not the other way around as stated in this sentence.
Reviewer 2 Report
The study evaluated prognostic factors (28-day mortality rate) in 210 patients suffering from nosocomial infections caused by Elizabethkingia species between 2005 and 2016.
Major comment.
Overall, the manuscript focuses on the difference in rifampicin MICs between surviving and non-surviving patients. However, this difference is minor and poorly significant. The authors advocate the use of ripampicin to treat Elizabethkingia infections, but none the patients of the study was treated with this antibiotic. On the other hand, other interesting differences (for example, the apparent higher efficacy of anti-pseudomonal penicillin plus glycopeptide combination) were not discussed.
Other comments
In the abstract section, please clarify the sentence lines 31-38. It would be clearer to indicate which variation of the different markers is associated with a higher survival rate. For example, "a low simplified acute physiology score II" rather than "the simplified acute physiology score II".
M&M
Line 93. Most nosocomial pneumonia (especially VAP) are caused by multiple pathogens. Was Elizabethkingia the only species infecting these patients. This point should be clarified. How did you evaluate the influence of the possibly associated microbial flora on patients’ survival?
Line 141. “… the MIC breakpoints for Staphylococcus spp. and Staphylococcus aureus were used to interpret the susceptibility of rifampin and vancomycin…”. MIC breakpoints determined for Gram-positive bacteria may not be adequate from Gram-negative ones such as Elizabethkingia. Therefore MICs cannot be interpreted as “susceptible” or “resistant” for the latter species.
RESULTS
Table 1. Please specify that the numbers in parenthesis represent percentages.
Line 173. “The survivors had … and a higher incidence of concomitant pathogens (80 [51.0%] vs. 17 [32.1%], p=0.017) than the non-survivors”. This finding merits to be discussed by the authors.
Table 3. It is difficult to understand why the range “0.008- >32” was used for rifampicin
Figure 1 could be deleted.
Line 199. “For the non-survivor group, the antibiotic susceptibility rate was found to be low for rifampin (19 [63.3%] vs. 73 [80.2%], p=0.060) and vancomycin (32 [23.9%] vs. 102 [95.3%], p=0.589) but no statistical significance was seen in case of moxifloxacin”. The results for these three antibiotics were not statistically significant.
Lie 202. “The MIC value of rifampin in the non-survivor group was significantly higher than that in the survivor group (11.9±15.6 vs. 6.0±11.7, p=0.031)”. This is a very low difference in MICs, corresponding only to twofold higher MICs for the non-survivor patients, which is usually considered not clinically significant. In both cases, MICs are higher than serum concentrations usually achieved for rifampicin.
Table 4. Anti-pseudomonal penicillin+glycopeptide: 12.7% survival vs 3.8% non-survival (p=0.045). This is a major finding of the study. The authors should discuss this finding and try to explain higher survival in patients receiving such antibiotic combination. As an example, glycopeptides that disrupt the bacterial membrane could favor penetration of beta-lactams into the bacteria resulting in higher activity.
Line 216. “In the multivariate logistic regression analysis, the MIC of rifampin was significantly associated with 28-day mortality (OR 1.045; 95% CI, 1.006-1.085; p=0.023). As the SAPS II increased, the mortality also increased (OR, 1.053; 95% CI, 1.022-1.084; p=0.001). The initial CRP/albumin ratio also had a significant association with 28-day mortality (OR, 1.030; 95% CI, 1.009-1.051; p=0.004)”. For all these parameters, the OR is close to one, which suggests that their association with better or worse survival is weak.
DISCUSSION
Line 223. “Recently, however, it has been found that E. meningoseptica is not only a human colonizer, but also an opportunistic pathogen causing in-hospital outbreaks”. This is not so recent considering that E. meningoseptica was formerly known as Chriseobacterium meningosepticum
Line 270. “An E-test for moxifloxacin demonstrated that it remained an active agent against all three Elizabethkingia spp. However, there were no significant differences in the mortality between groups that used moxifloxacin and those that did not. This might result from its low frequency of selection as a treatment regimen”. This sentence is unclear. Moxifloxacin as other FQ can rapidly select for resistant mutants in Gram-negative bacteria, which may lead to treatment failure.
Line 282. “It can thus be suggested that in patients showing severe Elizabethkingia infections, we can use vancomycin accompanied by rifampin unless the antibiotic susceptibility test of the causative species presents resistance to these drugs”. Data do not support this suggestion. None of the patients of the study was treated with rifampicin.
English editing is needed.
Round 2
Reviewer 2 Report
The manuscript has been much improved. Thank you to the authors to have answered all the comments I made for the first version of the manuscript. The relationship between rifampicin resistance and higher mortality rates remains curious since patients were not treated with this antibiotic. However, I would suggest the authors trying to make sense of this correlation by suggesting that RNA polymerase mutations causing resistance to rifampicin may have induced other virulence-related alterations. For example, it has been shown that Staphylococcus aureus strains resistant to rifampicin produced more biofilms.
